# Microstructure and Properties of As-Cast and Heat-Treated 2017A Aluminium Alloy Obtained from Scrap Recycling

**DOI:** 10.3390/ma14010089

**Published:** 2020-12-27

**Authors:** Mrówka-Nowotnik Grażyna, Kamil Gancarczyk, Andrzej Nowotnik, Kamil Dychtoń, Grzegorz Boczkal

**Affiliations:** 1Department of Material Science, Rzeszów University of Technology, Al. Powstańców Warszawy 12, 35-959 Rzeszów, Poland; KamilGancarczyk@prz.edu.pl (K.G.); nowotnik@prz.edu.pl (A.N.); kdychton@prz.edu.pl (K.D.); 2Faculty of Non-Ferrous Metals, AGH University of Science and Technology, Al. Mickiewicza 30, 30-059 Kraków, Poland; gboczkal@agh.edu.pl

**Keywords:** aluminium alloys, recycling, microstructure, heat treatment, mechanical properties, LM, SEM, DSC, XRD

## Abstract

The continuous increase in the consumption of aluminium and its alloys has led to an increase in the amount of aluminium scrap. Due to environmental protection, and to reduce the costs of manufacturing aluminum in recent years, a lot of research is devoted to recycling of aluminum alloys. The paper presents the results of research concerning the possibility of manufacturing standardized alloy 2017A from commercial and post-production scrap by continuous casting. Obtained from recycling process ingots were subjected to analysis of chemical composition and intermetallic phase composition. Based on the results of light microscopy (LM), scanning electron microscopy + electron dispersive spectroscopy (SEM + EDS), and X-ray diffraction (XRD) the following phases in the as-cast state were identified: θ-Al_2_Cu, β-Mg_2_Si, Al_7_Cu_2_Fe, Q-Al_4_Cu_2_Mg_8_Si_7_, and α-Al_15_(FeMn)_3_(SiCu)_2_. During solution heat treatment most of the primary precipitates of intermetallic phases, like θ-Al_2_Cu, β-Mg_2_Si, and Q-Al_4_Cu_2_Mg_8_Si_7_, were dissolved in the solid solution α-Al, and during natural and artificial aging they precipitate as strengthening phases θ-Al_2_Cu and Q-Al_4_Cu_2_Mg_8_Si_7_ with high dispersion. The highest hardness—150.3 HB—of 2017A alloy was obtained after solution heat treatment from 510 °C and aging at 175 °C. In the static tensile test the mechanical (R_m_ and R_p0_._2_) and plastic (A_5_) properties were determined for 2017A alloy in the cast state and after T4 heat treatment. The highest strength properties—tensile strength R_m_ = 450.5 MPa and yield strength R_0_._2_ = 268.7 MPa with good relative elongation A_5_ = 14.65%, were obtained after solution heat treatment at 510 °C/6 h/water quenching and natural aging at 25 °C for 70 h. The alloy manufactured from recycled scrap is characterized by relatively high mechanical properties.

## 1. Introduction

Aluminium and its alloys occupy a special place among light materials used in modern technology. The use of aluminium alloys is determined by both their prevalence in the earth’s crust and good physical, chemical, and mechanical properties, in particular low density and high relative strength (R_m_/δ), good electrical and thermal conductivity and corrosion resistance, as well as good technological properties [1,2,3,4,5,6,7,8]. It is not only the good mechanical properties that make aluminium and its alloys attractive, but also the fact that aluminium can be recycled indefinitely without loss of its properties, using 95 percent less energy than it takes to produce new aluminum. Almost 75% of Al ever produced is still in use nowadays as it can be recycled infinitely without compromising any of its exceptional properties or quality. Aluminium production, widely used for many applications, has one of the greatest energy alterations between primary and secondary production process at ~180 MJ/kg and ~15 MJ/kg, respectively [9,10]. This is why most manufacturers simply come to accept of expanding the usage of secondary materials, among other things—aluminium scrap [11,12]. Scrap became essential input material to the recycling process of aluminium alloys. New scrap, which is from production processes, arises during the production of aluminium semi-manufactured and final products. Old scrap, from postconsumer use, refers to those products which are out of service or are collected after disposal by clients. The amount of aluminium produced from old scrap has grown from one million tons in 1980 to 20 million tons in 2019. Recycling of old scraps need to apply more energy since they are often more contaminated than new scrap. The old scrap comes from vehicles (body construction, engine parts, cylinder heads, and electrical cabling), destroyed buildings (window frames and construction elements) and constructions, discarded packaging material (beverage cans), machinery equipment, as well as home and office appliances. Aluminium smelters are unable to safely accept this old scrap as its composition is usually unknown and it can be contaminated. Recycling aluminium scrap is divided into five steps: collecting, sorting, crushing remelting and casting. After collection from the scrap merchants, local, and regional authorities, scrap aluminium is then sorted. The sorting process divides scrap into coated and uncoated group together. Paper, polymer, and other types of non-aluminium recycling need to be removed from the sorting process. Sorted scrap is then crushed into small pieces by heavy machinery in order to reduce storage and handling costs. Uncoated scrap is loaded straightforwardly into a remelting furnace, where it is heated up and turned into liquid state. Coated scrap needs to be processed through a gas fired rotary furnace to remove the coatings, lacquers, paints etc. and then clean scrap are transferred to a large furnace—remelter. The molten aluminium scrap is usually cast at a temperature of 700 °C to form ingots. Along with the growing demand for environmental protection and green production, the recycling of Al alloys has become one of the most important production methods in the last few years, and is an important development direction in the aluminium industry. Therefore, in recent years, there has been a lot of work on the recycling of aluminium alloys [9,10,11,12,13,14,15,16,17,18,19], including the 2xxx group alloys [18,19,20,21]. Due to the unique combination of high strength properties, good fracture resistance, excellent fatigue properties, and tolerance to damage, 2xxx Al-Cu-Mg-(Si) aluminium alloys are ranked among the most important group of materials used mainly for highly loaded components of aircraft structures, automotive vehicles, rolling stock, and in construction [1,2,3,4,5,6,7,22,23,24,25,26]. The desirable properties of the 2xxx series alloys depend on their chemical composition and the heat treatment applied. The main alloying elements in commercial 2xxx series alloys are Cu, Mg, and small amounts of Si. In addition to the main alloying elements, the composition of the alloys contains elements from the group of transition metals, especially Mn, Fe, Cr, Ni, and Ti [1,2,3,4,5,6,7,26,27,28,29,30]. The solubility of these elements in solid aluminium is low, and therefore their presence even in insignificant amounts leads to the formation of intermetallic phases with the main alloying elements. Particles of intermetallic phases, formed during crystallization, mainly in eutectic reactions, are observed in the microstructure of the 2xxx series alloys. The presence of these phases in the microstructure of Al alloys is disadvantageous as they are brittle and break during plastic deformation. Moreover, they bind some of the main alloying elements such as Cu, Mg, and Si, thus reducing their participation in precipitation hardening. Two-component Al_2_Cu, Mg_2_Si phases; three-component Al_2_CuMg, Al_7_Cu_2_Fe, Al_6_(FeCu) phases and four-component particles of Al_4_CuMgSi_4_, Al_12_(FeMn)_3_Si, Al_12_(FeMn)_3_Si phases; and the five-component Al_15_(CuFeMn)_3_Si_2_ phase can be distinguished among the intermetallic phases occurring in the 2xxx series alloys [3,6,7,19,30,31]. The phase composition of the alloys, the morphology of the phase components of the microstructure, and hence the mechanical properties, can be changed in a wide range in heat treatment processes. Secondary intermetallic phases that are formed from the solution heat treated alloy during aging, which are the basis of precipitation hardening, have the greatest impact on increasing the mechanical properties of 2xxx group aluminium alloys. The type of strengthening phases and their volume fraction depend mainly on the chemical composition of the alloy and the percentage content of the elements forming these phases (Cu, Mg, and Si). The sequence of precipitation of strengthening phases from solution heat treated alloys and the type of phases present in the 2xxx Al-Cu-Mg-(Si) series alloys have been studied, characterized by various research techniques, such as TEM, SEM, XRD, and DSC, and presented in many scientific papers [32,33,34,35,36]. The analysis of the research results presented in the literature shows that depending on the content of Cu, Mg, and Si and the value of the Cu/Mg and Mg/Si ratio, 2xxx series aluminium alloys can be strengthened through five phases such as θ (Al_2_Cu), β (Mg_2_Si), S (Al_2_CuMg), Q (Cu_2_Mg_8_Si_6_Al_4_, Al_5_Cu_2_Mg_9_Si_7_, or Al_4_Cu_2_Mg_8_Si_7_), and precipitation of Si [37,38,39,40]. If the value of the Cu/Mg ratio is between 4 and 8, the strengthening takes place due to the precipitation of the θ (Al_2_Cu) and S (Al_2_CuMg) phase from the solution heat-treated alloy. However, the addition of Si to the alloy changes and greatly influences the precipitation sequence. The addition of Si in 2xxx series alloys promotes the precipitation of phases β (Mg_2_Si) and Q (Cu_2_Mg_8_Si_6_Al_4_) in addition to phase θ (Al_2_Cu). With a high content of Cu and with the Mg/Si > 1 ratio, phase β may also form in addition to phase θ. At an Mg/Si < 1 ratio, Q or S phases may be formed depending on the amount of Si. Very low Si content promotes the formation of the S phase, and a higher Si content promotes the formation of the Q phase [37,38,41].

This study involved the testing of a 2017A aluminium alloy for plastic processing obtained from scrap in a continuous casting process. Based on LM observation, SEM + EDS, and XRD diffractometric tests, the phase composition of the alloy after casting, solution heat treatment, and precipitation hardening was identified. In order to demonstrate the effect of heat treatment conditions on the mechanical properties of the investigated alloy, the precipitation hardening process was carried out using different values of solution heat treatment temperature and natural and artificial aging at 120 °C and 175 °C. The mechanical properties of the 2017A alloy obtained from scrap recycling were determined based on hardness measurement and static tensile test. The solution heat treatment and aging parameters were determined to ensure the best mechanical properties of the alloy while maintaining good plastic properties. Based on the results of DSC calorimetric and XRD diffractometric tests, the sequence of precipitation of strengthening phases from the solution heat-treated alloy obtained from scrap was determined.

## 2. Materials and Methods

The testing material was a 2017A aluminium alloy for plastic processing obtained from recycling. The ingot casting tests were carried out on a multi-strand continuous casting machine (The Foundry Research Institute, Cracow, Poland). The ingots were cast with the use of 4 oil-lubricated crystallizers (Figure 1). The input consisted of large pieces of scrap and production waste in the form of chips from the 2017A alloy. Scrap weighing ~265 kg was melted in a crucible resistance furnace (The Foundry Research Institute). Before casting the ingots from the liquid alloy, samples were taken and their chemical composition was checked on an ongoing basis, supplementing the alloying elements to the composition compliant with the PN EN 573-1 standard [42].

The chemical composition of the samples was analyzed using ARL-XTRa 3460 spectrometer (Thermo Electron SA, Ecublens, Switzerland) order to obtain a uniform, fine grain in the entire volume of the ingots, LSM’s TiBAl modifier/grain refiner (The Institute of Non-Ferrous Metals, Light Metals Division in Skawina, Poland) was introduced into the liquid alloy. Before casting, the liquid alloy was barbotage refining with the use of argon gas. The refining parameters were as follows; time 10 min, gas flow 10 L/min. After the conditions stabilized and the desired chemical composition was obtained, the casting of ingots was started using the parameters presented in Table 1.

### 2.1. Microscopic Tests

The microscopic examination of the 2017A alloy after casting and after heat treatment were performed using a Leica DMI 3000 M a Nikon Epiphot 300 light microscope (NIKON Solutions, Nagoya, Japan) and a HITACHI S-3400N scanning electron microscope (Hitachi High Technologies, Tokyo, Japan) with an EDS system for X-ray microanalysis (Oxford Instruments, High Wycombe, UK). The specimen size for the microscopic test was ~2.5 cm^2^. The specimens were cut from the ingot using a precision cutting machine Discotom–6 (Struers, Copenhagen, Denmark) and mounted in a bakelite (Struers, Copenhagen, Denmark). The specimen size for the microscopic test was ~2.5 cm^2^. The specimens were cut from the ingot using a precision cutting machine Discotom–6 and mounted in a bakelite. They were ground with SiC papers, 500, 800, 1000, and 1200 grit, and polished using 3 and 1 µm diamond polycrystalline suspensions. The final polishing was performed by application of alumina (Al_2_O_3_) suspension. Observation of the microstructure was carried out on polished, non-etched, and etched in the room temperature samples with a modified Keller’s reagent: 2 cm^3^ HF + 3 cm^3^ HCl + 20 cm^3^ HNO_3_ + 175 cm^3^ H_2_O. Fractographic studies of the fractures of the samples obtained after the static tensile test were also carried out using a scanning electron microscope (Hitachi High Technologies).

### 2.2. XRD Examination

The qualitative analysis of the phase components of the 2017A alloy microstructure after casting, solution heat treatment and natural aging was performed using on solid samples with a ground and polished surface. The phase composition of the 2017A alloy microstructure was identified using an X-ray diffractometer ARL XTRa from Thermo Fisher company (Thermo Electron SA, Ecublens, Switzerland). Filtered copper lamp (CuK_α,_λ = 0.1542 nm), with a voltage of 40 kV, current—30 mA, range 2θ = 20–80° and step size 0.02°/6 s was used. Phase composition was determined using the Powder Diffraction File (PDF) [43] developed and issued by the ICDD (The International Center for Diffraction Data).

### 2.3. Calorimetric Testing

In order to determine the melting point and measure the thermal effects related to phase transformations occurring in the scrap-cast 2017Aalloy, calorimetric tests were carried out using the differential scanning calorimetry (DSC). The tests were carried out using a NETZSCH STA 449F3 thermal analyzer (Netzsch, Selb, Germany) for cast and solution heat treated samples. During continuous heating of an ~30 mg cylindrical sample cut from an ingot at a rate of 10 °C/min from room temperature to 700 °C, the values of the eutectic dissolution temperature and the melting point of the 2017A alloy were determined. The values of the determined temperatures were very important while selecting the appropriate solution heat treatment temperature of the tested alloy. Additionally, in order to record the thermal effects related to the processes of precipitation of strengthening phases from the solution heat treated alloy, DSC tests were carried out on the samples immediately after solution heat treatment. Sample was continuously heated at a constant rate of 5 °C/min from the room temperature of 25 °C to 700 °C, using the measuring system to register the temperature and time of occurrence of thermal effects related to the separation of strengthening phases.

### 2.4. Heat Treatment

In order to increase the strength properties of the 2017A alloy obtained from scraps, the precipitation strengthening process was carried out. Solution heat treatment and aging processes allowing to obtain the best mechanical properties of the 2017A alloy, were carried out according to the parameters recommended in the standards requirements and literature [5,6,44]. The heat treatment conditions recommended for the 2017A alloy in the standards are as follows: solution heat treatment temperature is 495–505 °C and natural aging temperature is ~25 °C. There is not much data in the literature on the heat treatment of the tested alloy resulting in state T6. Therefore, the study investigated the influence of the T6 heat treatment parameters on the microstructure and hardness and compared them with the results obtained for the T4 heat treatment recommended in the literature for this alloy’s grade. The results of the calorimetric tests using the DSC method were very important when selecting the appropriate solution heat treatment temperature value for the 2017A alloy obtained from scrap in the continuous casting process. Exo- and endothermic peaks were registered on the DSC curve obtained during the heating of the alloy with a rate of 10 °C/min (Figure 2). When selecting the solution heat treatment temperature, two very distinct endothermic peaks were taken into account—the 1st with the peak maximum at 505 °C, which is most likely the result of the dissolution of the intermetallic phases occurring in the form of eutectics, and the 2nd with the peak maximum at 650 °C, which is the effect of dissolving a solid α-Al solution. The enlarged section of the DSC curve within the range of the occurrence of the 1st endothermic peak allowed for the precise selection of the solution heat treatment temperature value for the obtained 2017A ingots, so that the alloy did not melt during annealing. Moreover, it was established that the solution heat treatment temperature of 490 °C, recommended in the literature for this type of alloy, is ~10 °C before the 1st endothermic peak (Figure 2).

In the study, it was decided to carry out the precipitation strengthening process according to the diagram presented in Figure 3. In order to subject the samples to solute heat treatment process, they were heated in an electric resistance furnace to the temperature of the existence of a homogeneous solid α-Al solution, i.e., 490 °C, 500 °C, and 510 °C, annealed for 6 h and cooled in water to ~15–20 °C to solutionize. Throughout the annealing of the samples, the temperature value was recorded using a thermocouple, which remained at the set level of 490 °C, 500 °C, and 510 °C ± 3 °C. Natural aging of the alloy was carried out in the air at ~25 °C for 200 h to obtain the T4 state, while artificial aging was carried out at 120 °C and 175 °C for 145 h—obtaining the T6 state of the alloy. After artificial aging, the samples were cooled in water and their HB hardness was measured.

### 2.5. Mechanical Properties

The studies on mechanical properties included hardness measurement and a static tensile test. The hardness of the alloy was measured using an Instron Wolpert hardness tester (Instron, High Wycombe, UK) using the HBW2.5/62.5 Brinell method (62.5 kg load, d = 2.5 mm indenter). First, the hardness of the 2017A alloy ingots obtained from scrap using the method of continuous casting was measured. Hardness measurements were carried out on three samples cut out from the each ingots. All the given values are the average of three measurements for each specimen.

Hardness measurement was also carried out for samples subjected to cooling from different solution heat treatment temperatures of 490 °C, 500 °C, and 510 °C and during natural and artificial aging of samples with different temperatures (25 °C, 120 °C, and 175 °C) and aging time (from 0.5 to 200 h).

The static tensile test was carried out for the 2017A alloy in as-cast state and after T4 heat treatment (solution heat treated from 490 and 510 °C/6 h/water cooling + natural aging for 4, 24, 48, and 70 h) using an INSTRON 8801 universal testing machine (Instron) in accordance with the PN-EN 10002-1:2004 standard [45]. Round fivefold test pieces with a diameter of 10 mm were used (Figure 4). For ingots of 2017A alloy in as-cast state and after solution heat treatment process and after aging process with application of different time, three samples were taken from each ingot and then they were subjected to tensile tests. Apparent yield strength R_0_._2_, tensile strength R_m_, relative elongation A_5_, and reduction Z were determined.

## 3. Results and Discussion

Four round ingots with a diameter of 70 mm and a length of 6020 mm were obtained in the continuous casting process. The outer surface of the ingots was free from cracks, blows, or other casting defects (Figure 5a). Samples were cut out of the obtained ingots and subjected to spectrometric analysis in order to check the exact chemical composition. The obtained results indicate that the content of the alloying elements is within the ranges required for this alloy grade (Table 2). The observation of the macro- and microstructure of the 2017A alloy ingots obtained from scrap using the method of continuous casting was also carried out. The condition for obtaining high strength properties of cast alloys is to obtain a compact, fine-grained microstructure devoid of internal discontinuities. The observation revealed that the ingots obtained in the study did not show discontinuities and casting defects and were characterized by a fine-grained, homogeneous macrostructure and microstructure on the ingot cross section (Figure 5b). Such microstructure ensures good mechanical properties of the tested alloy already after casting. The average hardness of 2017A alloy ingots made from scrap in the continuous casting process was 98 HB and compared to the value of 58 HB of conventionally cast ingots is relatively high. The mechanical properties of the investigated alloy after casting, determined in the static tensile test, are as follows: tensile strength R_m_ = 315.7 MPa, yield strength R_0_._2_ = 165.2 MPa, and relative elongation A_5_ = 8.8%. The mechanical properties of 2017 A alloy produced by melting pure elements (not by recycling) determined in the static tensile test, are as follows: tensile strength R_m_ = 250.7 MPa, yield strength R_0_._2_ = 125.2 MPa and relative elongation A_5_ = 10.0% [4,5,6].

The observation of the microstructure of a sample cut from the 2017A alloy ingot obtained in the continuous casting process is shown in Figure 6. The microstructure of the alloy after casting is fine-grained and highly homogeneous. It is typical for the cast state and consists of α-Al solid solution dendrites and precipitates of intermetallic phases crystallizing mainly in the form of eutectics distributed in interdendritic regions. The main alloying elements—Cu, Mg, and Si—partially dissolved in the α-Al solid solution—strengthening it by the solution, and partially formed intermetallic phases with aluminium or between themselves and the elements of transition metals: Fe and Mn (Figure 6).

Observation of the sample carried out under high magnification on the SEM scanning microscope showed that the primary precipitates formed in the crystallization process differ in morphology (“Chinese characters”) plaques, spheroids, polyhedrons and needles), and color (Figure 6 and Figure 7). In order to accurately identify the phase components of the microstructure of the tested alloy after casting, a SEM observation combined with the point + EDS microanalysis (Figure 7 and Table 3) and XRD diffraction tests (Figure 8) were performed. The results of the obtained tests showed that in the phase composition of the tested alloy after casting obtained by recycling, there are precipitates of the two-component phases—θ-Al_2_Cu and β-Mg_2_Si, a three-component Al_7_Cu_2_Fe phase, a four-component Q-Al_4_Cu_2_Mg_8_Si_7_ phase, and a five-component α-Al_15_(FeMn)_3_(SiCu)_2_ phase. The microscopic observation and the analysis of the XRD spectrum show that the higher relative volume, apart from the α-Al solid solution, is occupied by the precipitates of the two-component θ-Al_2_Cu phase. The XRD spectrum of the examined alloy after casting showed the largest number of high-intensity reflections from this phase (Figure 8).

The mechanical properties of the tested 2017A alloy after casting are good (98 HB, R_m_ = 315.7 MPa, R_0_._2_ = 165.2 MPa, and A_5_ = 8.8%); however, they can be significantly increased by subjecting it to precipitation hardening. In order to subject the examined alloy to solution heat treatment process, the samples were subjected to annealing at 490, 500, and 510 °C for 6 h and cooling in water. Observation of the samples solution heat treated at 490 and 510 °C and naturally aged showed that during annealing for 6 h, most of the primary precipitates of intermetallic phases observed after casting (Figure 6, Figure 7, Figure 8) were dissolved in the solid α-Al solution (Figure 9). The morphology of the remaining undissolved precipitates of intermetallic phases, mainly those containing Fe, was also changed. No acicular or lamellar precipitates of the Al_7_Cu_2_Fe phase were visible in the cast state (Figure 6 and Figure 7). Based on own research [46] and literature data [1,3,28,30], it can be assumed that these precipitates were transformed into α-Al_15_(FeMn)_3_(SiCu)_2_ phase particles as a result of annealing to solution heat treatment at 490 °C and 510 °C. It was also found that the sample solution heat treated from the highest temperature, i.e., 510 °C (Figure 9b), was characterized by a smaller relative volume of primary intermetallic phase precipitates compared to the sample solution heat treated from 490 °C (Figure 9a). The use of a higher annealing temperature increases the solubility of the primary particles of intermetallic precipitates, mainly θ-Al_2_Cu, β-Mg_2_Si, and Q-Al_4_Cu_2_Mg_8_Si_7_, in the α-Al solid solution. During aging of the alloy, the particles of these phases are precipitated in the form of the following strengthening phases: secondary, dispersive and evenly distributed throughout the entire volume of the alloy (Figure 9).

The diffractometric studies of the solution heat treated sample from the 2017A alloy confirm the microscopic observation (Figure 10). The intensity of the peaks originating from intermetallic phases in the XRD spectrum of the solution heat treated sample is much smaller compared to the cast state (Figure 8). The analysis of the diffraction pattern of the solution heat treated 2017A alloy showed that during annealing of this alloy, most of the precipitates of the two-component phases θ-Al_2_Cu and β-Mg_2_Si and the four-component Q-Al_4_Cu_2_Mg_8_Si_7_ phase dissolved. Moreover, only one peak from the particles of the three-component Al_7_Cu_2_ phase was found, which confirms microscopic observations and may indicate that during the annealing, most of the Al_7_Cu_2_Fe phase precipitates were converted into the α-Al_15_(FeMn)_3_(SiCu)_2_ phase.

Measurement of the hardness of samples from the 2017A alloy that differ in the value of the solution heat treatment temperature showed that the hardness of the alloy increases with the increase in solution heat treatment temperature (Figure 11). The sample solution heat treated from 490 °C is characterized by the lowest hardness value, i.e., 81.5 HB, while the sample solution heat treated from 510 °C is the highest, i.e., 91.6 HB (Figure 11). The highest hardness of the alloy after solution heat treatment at the highest temperature, i.e., 510 °C, results from the almost complete dissolution of intermetallic phases in the solid solution (Figure 9b), which causes homogenization and strengthening of the solution heat treated alloy.

The further growth of the alloy’s hardness was obtained during natural and artificial aging. For this purpose, immediately after solution heat treatment, the samples were subjected to natural aging at room temperature of ~25 °C for 200 h, obtaining the T4 state, and artificial aging at 120 °C and 175 °C for 145 h, obtaining the T6 state. In order to demonstrate the effect of solution heat treatment temperature and aging time on the change in the properties of the tested alloy, the HB hardness measurement of the samples after different natural aging times was carried out. The obtained results were used to draw aging curves HBW2.5/62.5 = f(t) (T = const) (Figure 12). It was found that the solution heat treatment temperature affects the hardness of the naturally aged aluminium alloy 2017A. The tested alloy, after solution heat treatment from the lowest temperature of 490 °C, has the lowest value of maximum hardness—123.5 HB—obtained during natural aging (Figure 12). The maximum hardness of samples solutionized from 500 °C and 510 °C and naturally aged is comparable and amounts to 127.6 HB and 128.3 HB respectively (Figure 12). Regardless of the applied solution heat treatment temperature, the alloy, after reaching the maximum hardness value during natural aging, does not become overaged, characterized by a reduction in hardness. Hardness close to the maximum values remains at a similar level during natural aging for 200 h (Figure 12). The low temperature of aging does not lead to the degradation of the microstructure demonstrated by the growth and coagulation of strengthening phase particles and, consequently, deterioration of strength properties.

Moreover, it was found that the solution heat treatment temperature also influences the kinetics of precipitation of strengthening phases during natural aging of the solution heat-treated 2017A aluminium alloy. The maximum hardness value—128.3 HB—after solution heat treatment from the highest temperature of 510 °C was obtained already after ~48 h of natural aging. A comparable hardness value was obtained for the alloy solutionized from 500 °C after ~96 h of aging. The longest time—180 h—to reach the maximum hardness value of 123.5 HB was recorded for the alloy solution heat treated from the lowest temperature of 490 °C (Figure 13).

Based on the obtained aging curves—HBW2.5/62.5 = f(t) (T = const) (Figure 12)—and the data presented in Diagram 13, it is also possible to determine the influence of the solution heat treatment temperature on the rate of hardness increase to the maximum value by calculating, e.g., gradients of hardness grad ΔHB/Δt, using the following equations.
grad HB = ΔHB/Δt(1)
ΔHB = HB_max_ − HB_w_(2)
where
-HB_max_—maximum hardness value obtained during aging.-HB_w_—starting hardness of the alloy (directly after homogenization and solution heat treatment).-Δt—aging time after which the alloy reached maximum hardness value HB_max_.

The values of parameters determined from aging curves HBW2.5/62.5 = f(t); T = const and hardness gradient calculated from Equation (1) are presented in Table 4.

Based on the obtained hardness gradient values, it can be concluded that the rate of hardness increase to the maximum value during natural aging increases with the increase in the solution heat treatment temperature. The alloy solutionized from the temperature of 490 °C (HB grad = 0.233 HB/h) is characterized by the lowest rate of hardness increase to the maximum value, and the alloy solution heat treated from the temperature of 510 °C (HB grad = 0.764 HB/h) is characterized by the highest value (Table 4 and Figure 14).

For the identification of the type of strengthening phases and to determine the sequence of strengthening phases during natural aging, the diffractometric XRD and the calorimetric tests were performed (Figure 15 and Figure 16). As is known, the processes of precipitation hardening of aluminium alloys of the 2xxx series have been the subject of research conducted in R&D centers around the world for many years [23,26,32,33,34,35,36,37,38,39,40,41]. As indicated by the authors of studies [36,37,38,39,40], the content of the main alloying elements Cu, Mg, and Si, and the values of the Cu/Mg and Mg/Si ratios affect the precipitation sequence and the type of strengthening phases in the 2xxx series alloys. Alloys of this group are strengthened mainly by precipitation of the following phases from solution heat treated alloys: θ (Al_2_Cu), β (Mg_2_Si), S (Al_2_CuMg), and Q (Cu_2_Mg_8_Si_6_Al_4_, Al_5_Cu_2_Mg_9_Si_7_, or Al_4_Cu_2_Mg_8_Si_7_). If the value of the Cu/Mg ratio is between 4 and 8, the strengthening takes place due to the precipitation of mainly the θ (Al_2_Cu) and S (Al_2_CuMg) phase from the solution heat-treated alloy. However, the addition of Si to the alloy changes the precipitation sequence. Low Si content promotes the formation of the S phase, and a higher Si content promotes the formation of the Q phase. The addition of Si in 2xxx series alloys promotes the precipitation of phases β (Mg_2_Si) and Q (Cu_2_Mg_8_Si_6_Al_4_) in addition to phase θ (Al_2_Cu). If the value of the Mg/Si ratio is < 1.7, the 2xxx series alloys are strengthened mainly by phases θ (Al_2_Cu) and Q Cu_2_Mg_8_Si_6_Al_4_. In the studied 2017A alloy from recycling, the content of Cu is 4.01% and the content of Si is quite high and amounts to 0.49%. The value of the Cu/Mg ratio is 5.57, and for Mg/Si it is 1.46. Thus, it can be concluded that the investigated alloy will be strengthened mainly by dispersive particles of strengthening phases θ-Al_2_Cu and Q-Al_4_Cu_2_Mg_8_Si_7_. The results of XRD diffraction tests obtained for the 2017A alloy, subjected to solution heat treatment and naturally aged for 200 h, confirm that the strengthening occurs mainly as a result of the precipitation of these two phases (Figure 15). The XRD spectrum obtained for the solution heat treated alloy (Figure 10) showed that most of the precipitates had dissolved in the α-Al solid solution matrix. The XRD spectrum (Figure 15) of the solution heat treated and naturally aged for 200 h alloy showed an increase in the intensity of the peaks, mainly from the strengthening particles of phases θ-Alu and Q-Al_4_Cu_2_Mg_8_Si_7_.

In order to determine the sequence of the strengthening phase precipitation, a sample from the 2017A alloy was heated in a calorimeter immediately after solution heat treatment at a constant rate of 5 °C/min from room temperature to 700 °C. Exo- and endothermic peaks were observed on the DSC curve during heating of the solution heat-treated alloy (Figure 16). Exothermic peaks are marked with numbers from 1 to 6, while endothermic peaks with capital letters A–G. The section of the DSC curve in the temperature range of 50 °C to 400 °C in which the process of strengthening phase precipitation was recorded during continuous heating of the solution heat treated alloy was enlarged. Based on the results of XRD diffractometric (Figure 15) and DSC calorimetric tests (Figure 16), it can be concluded that the main strengthening phases in the tested 2017A alloy will be the precipitates of phases θ-Al_2_Cu and Q-Al_4_Cu_2_Mg_8_Si_7_. Based on the DSC curve obtained during continuous heating of the solution heat treated 2017 alloy (Figure 16), it should be assumed that the precipitation process begins with the formation of clusters of Cu and Mg atoms—exothermic peak 1 with a maximum at 70 °C (Table 5). The increase in temperature leads to the formation of GP and GPB zones coherent with the matrix (exothermic peaks 2 and 3) and then to their dissolution (endothermic peaks A and B). The highest intensity exothermic peaks 4 and 5 result from the precipitation of the θ”, θ’ and Q’, Q” phases, partially coherent with the matrix, from the solution heat treated alloy. The precipitation of these phases results in obtaining the maximum mechanical properties of the alloy. Further heating leads to dissolution (endothermic peak D) of metastable phases θ’ and Q’ and the formation of stable equilibrium precipitates of phases θ-Al_2_Cu and Q-Al_4_Cu_2_Mg_8_Si_7_ incoherent with the matrix.

The study also covered the effect of the temperature of solution heat treatment and artificial aging on the hardness of the 201 alloy. As, during natural aging, similar hardness values were obtained for the samples solution heat treated from 500 °C and 510 °C, it was decided that two extreme values of the solution heat treatment temperature, i.e., 490 °C and 510 °C would be used. Based on hardness measurement of solution heat treated and artificially aged samples, it was established that the solution heat treatment temperature influences the hardness of the 2017A aluminium alloy, regardless of the applied artificial aging temperature (Figure 17). The 2017A aluminium alloy, solution heat treated from 510 °C and artificially aged both at 120 °C and 175 °C, obtains higher hardness values during aging.

Based on the analysis of the results of hardness measurement obtained during natural and artificial aging (Figure 18), it was found that the sample solutions heat treated from 510 °C show the highest hardness during both natural aging at ~20 °C (128.3 HB) and artificial aging at 120 °C (143 HB) and 175 °C (150.3 HB). Samples aged at the highest temperature, i.e., 175 °C, regardless of the applied solution heat treatment temperature (490 °C and 510 °C), are characterized by the highest hardness values (Figure 17, Figure 18, Figure 19). Aging at 120 °C leads to a hardness very similar to that obtained during natural aging. The tested alloy is characterized by the lowest hardness value of 123.5 HB after solution heat treatment from the lowest temperature of 490 °C. The maximum hardness of samples solution heat treated from 500 °C and 510 °C and naturally aged is comparable and amounts to 127.6 HB and 128.3 HB respectively (Figure 12).

For samples solution heat treated from 490 °C and 510 °C and naturally and artificially aged, the ΔHB/Δt hardness gradients were also calculated, characterizing the rate of hardness increase of the 2017A alloy to the maximum value depending on the solution heat treatment and aging temperature (Table 6).

Based on the obtained results, it can be concluded that only in the case of natural aging does the solution heat treatment temperature affect the rate of hardness increase up to the maximum value. The grad HB value for the alloy solution heat treated from 490 °C and aged naturally is 0.233 HB/h, and for the alloy solution heat treated from 510 °C and aged naturally it is more than twice as high and amounts to 0.746 HB/h. In the case of artificial aging at 120 °C, the rate of hardness increase to the maximum value is almost the same (490 °C—grad HB = 0.496 HB/h, 510 °C—grad HB = 0.475 HB/h). On the other hand, in the case of artificial aging at 175 °C, the rate of hardness increase to the maximum value is slightly higher for the alloy after solution heat treatment at 490 °C (490 °C—grad HB = 2.05 HB/h, 510 °C—grad HB = 1.946 HB/h) (Table 6, Figure 20).

In order to demonstrate the effect of natural aging on the mechanical properties of the recycled 2017A alloy, a static tensile test was performed on samples solution heat treated from 490 °C and 510 °C, differing in the rate of natural aging (Table 7). After the static tensile test, similar relationships were observed as in the case of hardness. The obtained results indicate that both tensile strength R_m_ and apparent yield strength R_0_._2_ increase with the increase of the natural aging time. The highest strength properties for both solution heat treatment temperature values were achieved after 70 h of natural aging. As in the case of hardness, higher strength properties (Table 7, Figure 21a,b) of the tested alloy while maintaining good plastic properties (Table 7, Figure 21c) were obtained for samples solution heat treated from 510 °C.

SEM observation of the samples subjected to the static tensile test, differing in the solution heat treatment temperature and the natural aging time, was also carried out (Figure 22). The observation showed that regardless of the parameters of the heat treatment, the fractures showed features characteristic of the morphology of a ductile fracture—there were cavities and voids on fracture surfaces. It was found that intergranullar cracking is dominant in the 2017 alloy, regardless of the applied heat treatment parameters. The crack develops through nucleation, growth, and void connection. Particles of secondary strengthening phases θ-Al_2_Cu and Q-Al_4_Cu_2_Mg_8_Si_7_ and undissolved primary intermetallic phases θ-Al_2_Cu, Al_7_Cu_2_Fe and α-Al_15_(FeMn)_3_(SiCu)_2_ also play a significant role in the cracking process of the investigated alloy. The primary void nucleation sites are the particles of precipitates of strengthening phases θ-Al_2_Cu and Q-Al_4_Cu_2_Mg_8_Si_7_.

It was also found that in the areas where the primary undissolved precipitates of the θ-Al_2_Cu phase occurred, the decohesion process is initiated at the matrix/particle interface (Figure 23a). Fractographic studies also showed cracking of hard, brittle primary particles of iron-containing phases: Al_7_Cu_2_Fe and α-Al_15_(FeMn)_3_(SiCu)_2_. These particles are defragmented as a result of tensile loads (Figure 23b).

## 4. Conclusions

The 2017 alloy, obtained in the process of scrap recycling, is characterized by homogeneous and fine grains throughout the entire volume of the ingot. No discontinuities or casting defects were observed on the cross section of the ingot. The mechanical properties of the as-cast ingot obtained using the continuous casting method are as follows; tensile strength R_m_ = 315.7 MPa, yield strength R_0_._2_ = 165.2 MPa, and relative elongation A_5_ = 8.8%.The microstructure of the as-cast 2017 alloy consists of precipitates of intermetallic phases crystallizing mainly in the form of eutectics in the interdendritic regions of the solid α-Al solution. The precipitates of the following phases were identified based on the results of LM, SEM + EDS and XRD studies: θ-Al_2_Cu, β-Mg_2_Si, Al_7_Cu_2_Fe, Q-Al_4_Cu_2_Mg_8_Si_7_, and α-Al_15_(FeMn)_3_(SiCu)_2_.The value of solution heat treatment temperature affects the mechanical properties and kinetics of precipitation during the natural aging of the 2017 alloy. The highest hardness and the rate of hardness increase to the maximum value were obtained for the 2017 aluminium alloy solution heat treated at 510 °C and aged naturally.The solution heat treatment temperature also affects the hardness of the studied alloy during artificial aging. It was found that artificial aging allows for achieving higher hardness values compared to natural aging. Alloy 2017 solution heat treated from 510 °C and aged at 175 °C shows the highest hardness. During aging at 120 °C, the hardness is close to the value achieved during natural aging. It was found that during artificial aging, the solution heat treatment temperature does not affect the rate of hardness increase to the maximum value.Strength properties and hardness for both solution heat treatment temperatures—500 °C and 510 °C—increase with the increase in aging time. The highest strength properties—tensile strength R_m_ = 450.5 MPa and yield strength R_0_._2_ = 268.7 MPa—while maintaining good plastic properties, relative elongation A_5_ = 14.65%, were obtained after annealing at 510 °C/6h and solution heat treatment in water and natural aging at 25 °C for 70 h.Based on calorimetric and diffractometric studies, it was established that the process of precipitation of strengthening phases from a solution heat treated alloy proceeds according to the following pattern: GP/GPB zones→ θ/Q” → θ/Q” → θ-Al_2_Cu/Q-Al_4_Cu_2_Mg_8_Si_7_. The maximum strengthening of the alloy is the results of the precipitation of metastable, transitional phases θ” and θ’ and Q” and Q’.Based on fractographic studies, it was established that in the tested alloy, regardless of the parameters of heat treatment, decohesion under tensile stress occurs through nucleation, growth, and void connection. It was also found that in the places where the primary undissolved precipitates of the θ-Al_2_Cu phase occurred, the decohesion process was initiated at the matrix/particle interface. The particles of the primary hard and brittle phases containing iron: Al_7_Cu_2_Feand Al_15_(FeMn)_3_(SiCu)_2_ are defragmented as a result of tensile loads.

## Figures and Tables

**Figure 1 materials-14-00089-f001:**
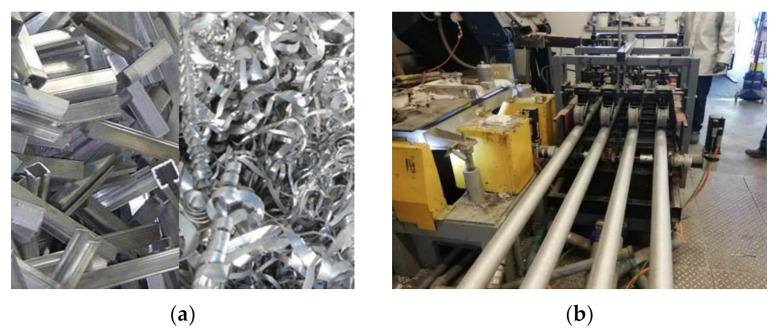
Scrap constituting the input for melting and obtaining a technical 2017A alloy (**a**) and a station for continuous casting of ingots (**b**).

**Figure 2 materials-14-00089-f002:**
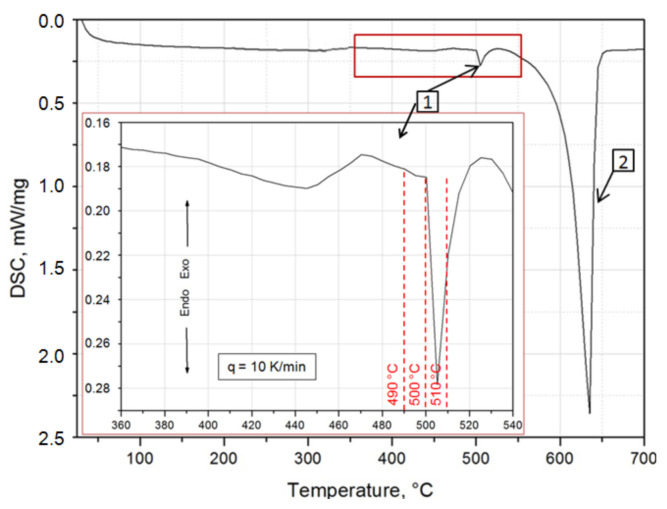
Thermogram (DSC) obtained during the heating of the 2017A alloy from room temperature to 700 °C at a rate of 10 °C/min with an enlarged section of the thermogram in the temperature range of 360 °C to 540 °C. The peak denoted as (1) is the result of the dissolution of the intermetallic phases occurring in the form of eutectics, and the peak dented as (2) is the effect of dissolving a solid α-Al solution.

**Figure 3 materials-14-00089-f003:**
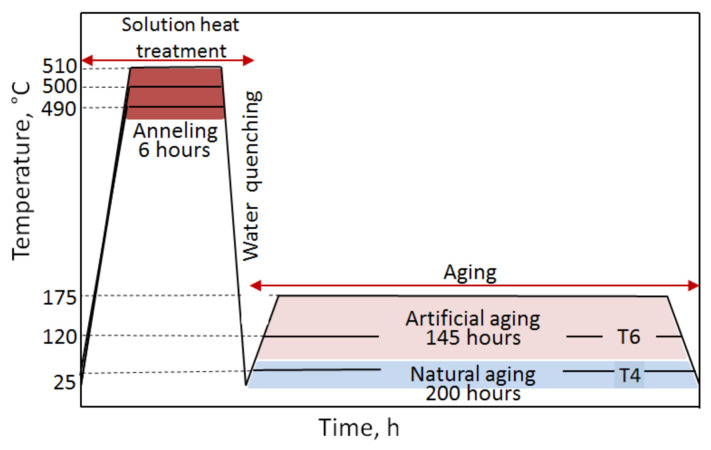
Diagram presenting heat treatment of the 2017A aluminium alloy.

**Figure 4 materials-14-00089-f004:**
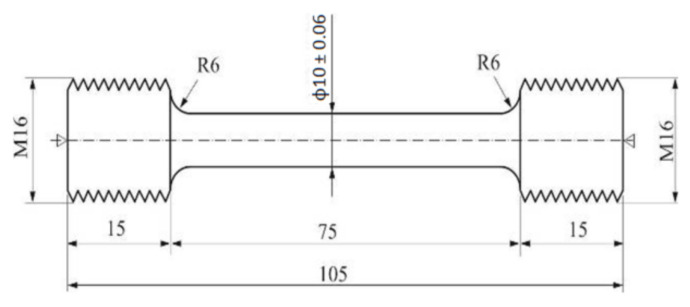
Shape and size of samples for static tensile test.

**Figure 5 materials-14-00089-f005:**
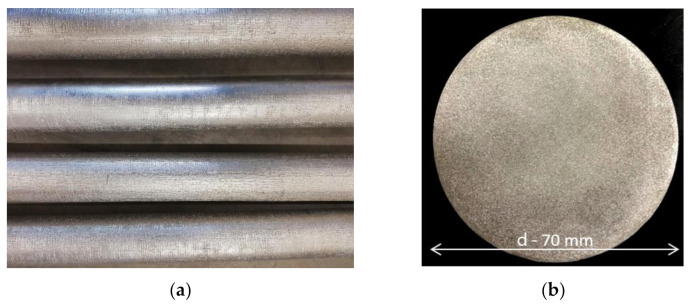
Outer surface of the obtained ingots (**a**) and a typical macrostructure on the ingot cross section (**b**).

**Figure 6 materials-14-00089-f006:**
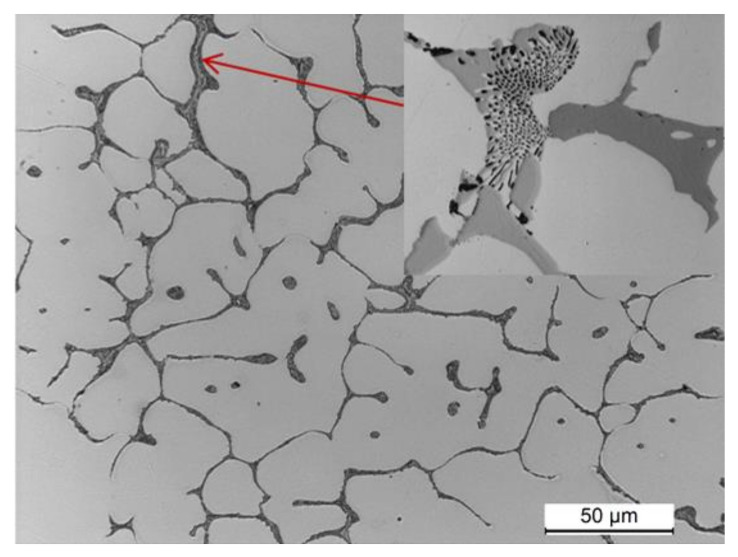
LM microstructure of the 2017A alloy obtained from recycling after casting.

**Figure 7 materials-14-00089-f007:**
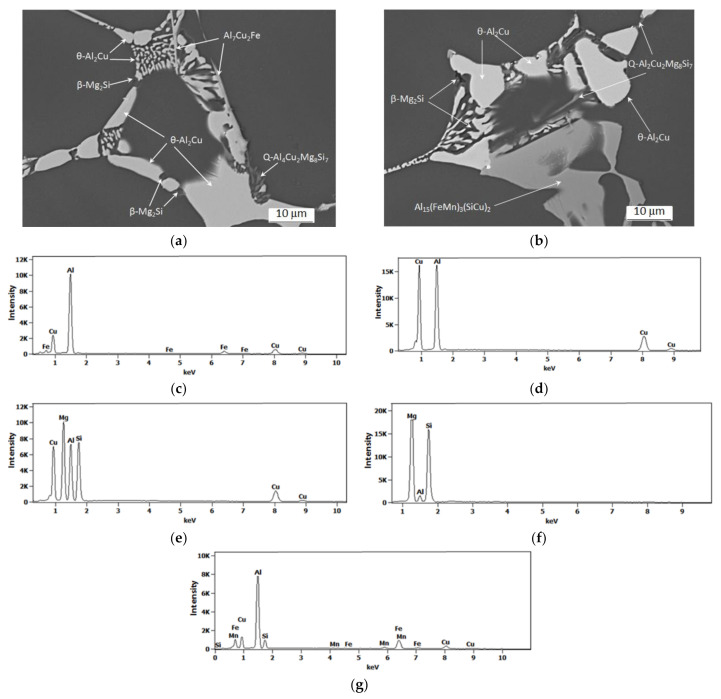
SEM microstructure of the 2017A alloy obtained from recycling after casting (**a**,**b**) and typical EDS spectra of the analyzed particles: (**c**) Al_7_Cu_2_Fe phase, (**d**) θ-Al_2_Cu phase, (**e**) Q-Al_4_Cu_2_Mg_8_Si_7_ phase, (**f**) β-Mg_2_Si phase and (**g**) α-Al_15_(FeMn)_3_(SiCu)_2_ phase.

**Figure 8 materials-14-00089-f008:**
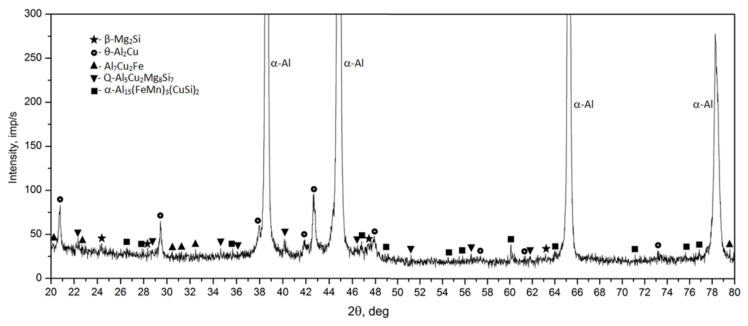
X-ray diffraction pattern of the 2017A alloy after casting.

**Figure 9 materials-14-00089-f009:**
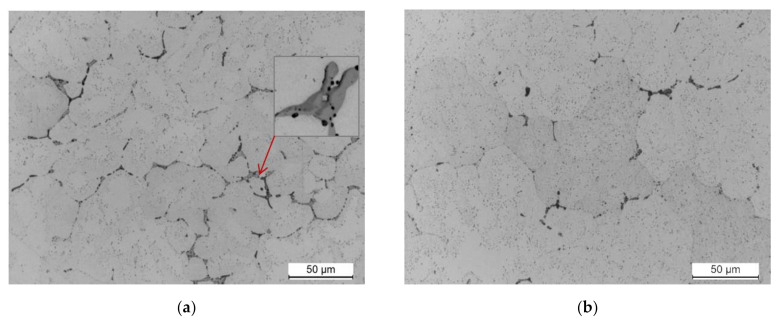
Microstructure of the 2017A alloy after solution heat treatment from (**a**) 490 °C and (**b**) 510 °C and natural aging.

**Figure 10 materials-14-00089-f010:**
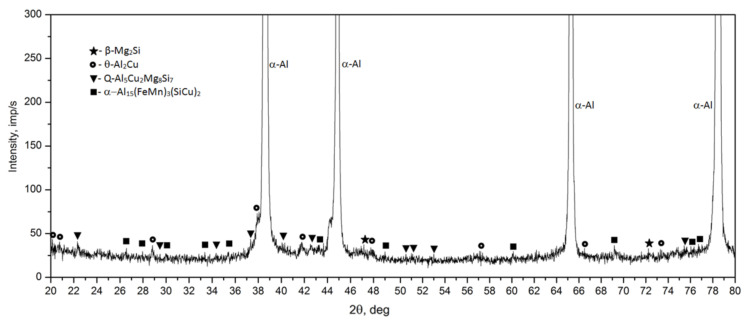
X-ray diffraction pattern of the 2017A alloy after solution heat treatment.

**Figure 11 materials-14-00089-f011:**
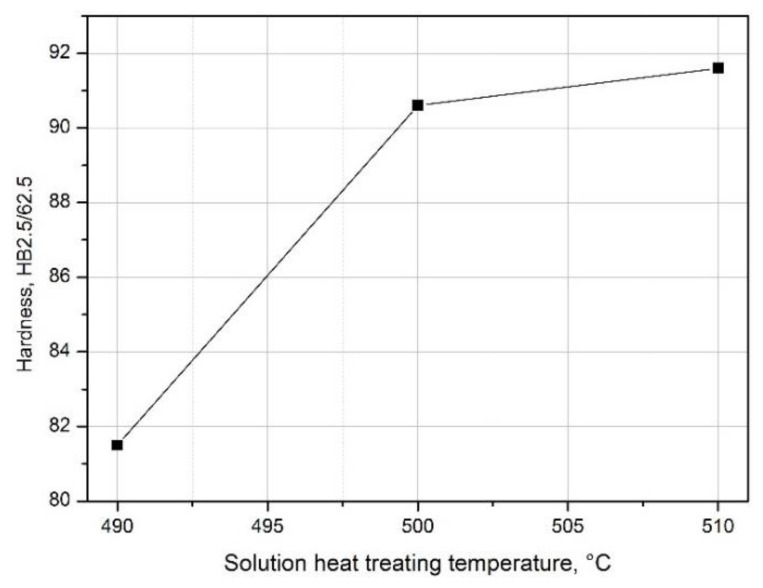
Influence of solution heat treatment temperature on the 2017A alloy hardness.

**Figure 12 materials-14-00089-f012:**
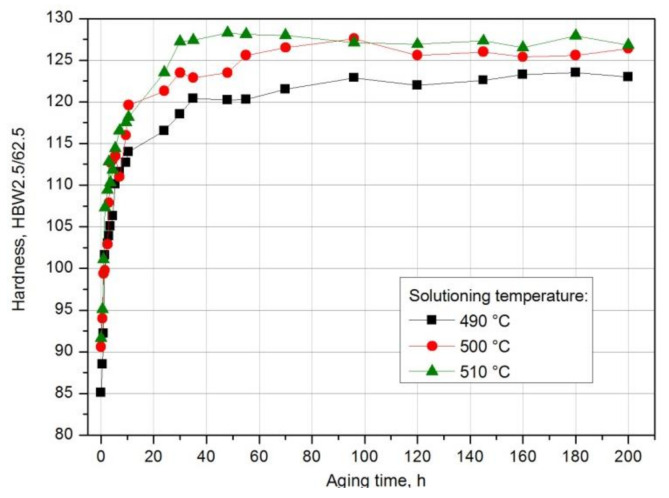
Effect of solution heat treatment temperature on the change in hardness of a naturally aged alloy.

**Figure 13 materials-14-00089-f013:**
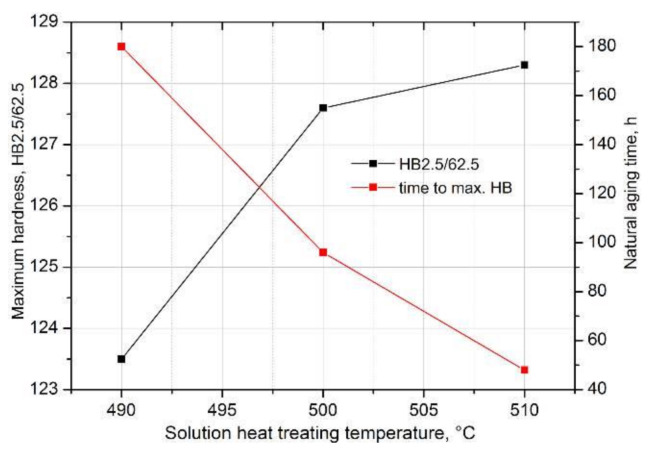
Influence of solution heat treatment temperature on the value and time to obtain the maximum hardness for the 2017A aluminium alloy naturally aged at ~25 °C.

**Figure 14 materials-14-00089-f014:**
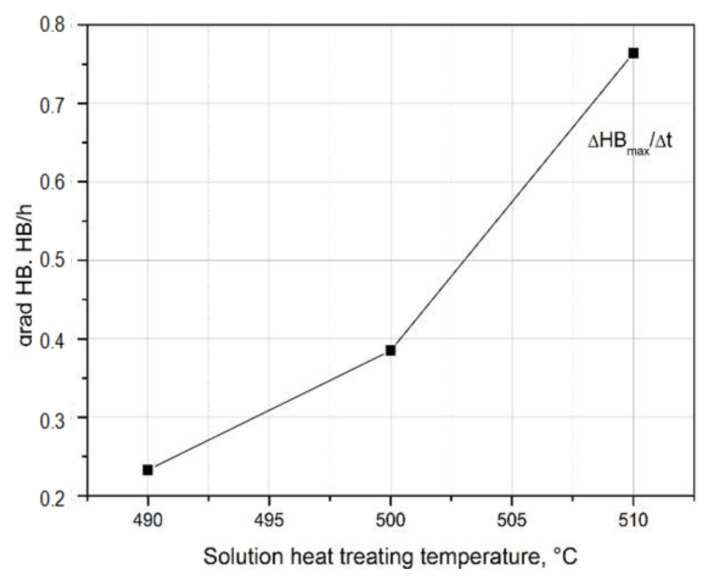
Influence of solution heat treatment temperature on the rate of hardness increase to the maximum value of the 2017A alloy obtained during natural aging.

**Figure 15 materials-14-00089-f015:**
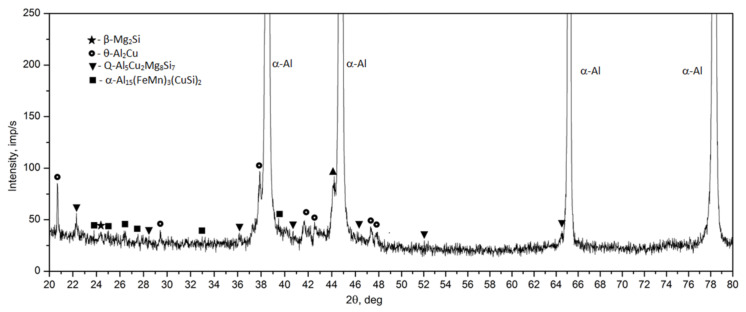
Diffraction pattern of the 2017A alloy solution heat treated from 510 °C and aged naturally for 200 h.

**Figure 16 materials-14-00089-f016:**
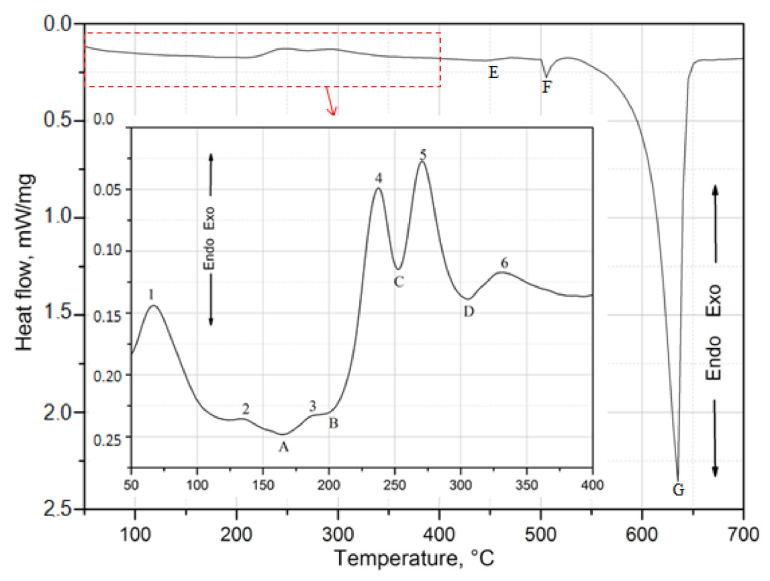
DSC thermogram of solution heat treated 2017 alloy, heated up to 700 °C with a heating rate of 10 K/min.

**Figure 17 materials-14-00089-f017:**
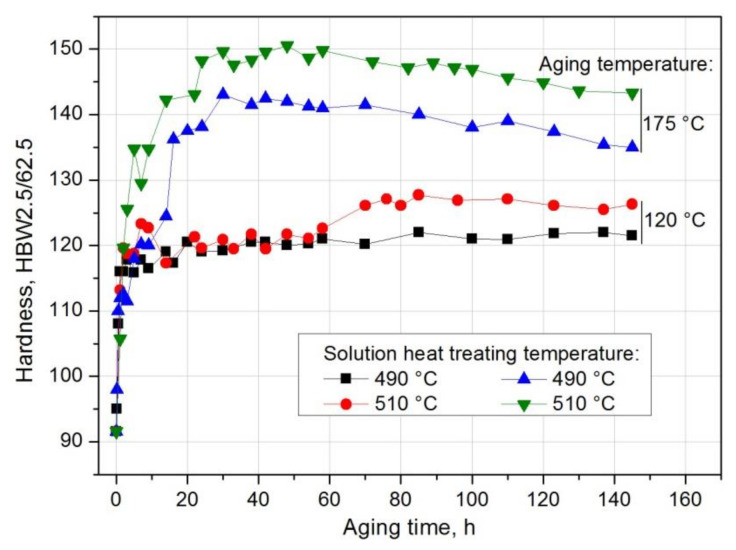
Effect of solution heat treatment temperature (490 °C and 510 °C) and time of artificial aging at 120 °C and 175 °C on the hardness of the EN AW-2017A aluminium alloy.

**Figure 18 materials-14-00089-f018:**
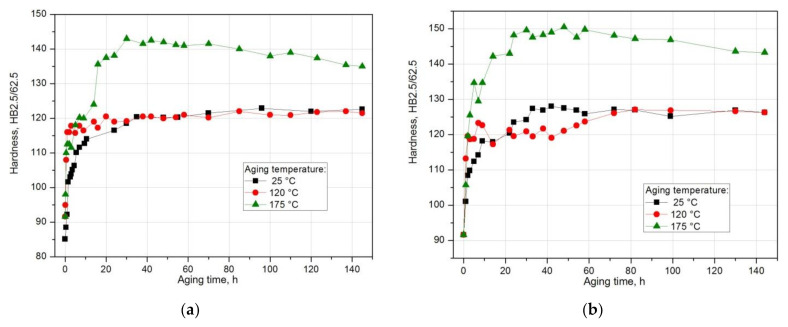
Effect of aging temperature and time on the change in the hardness of an alloy solution heat treated from (**a**) 490 °C and (**b**) 510 °C and aged naturally and artificially.

**Figure 19 materials-14-00089-f019:**
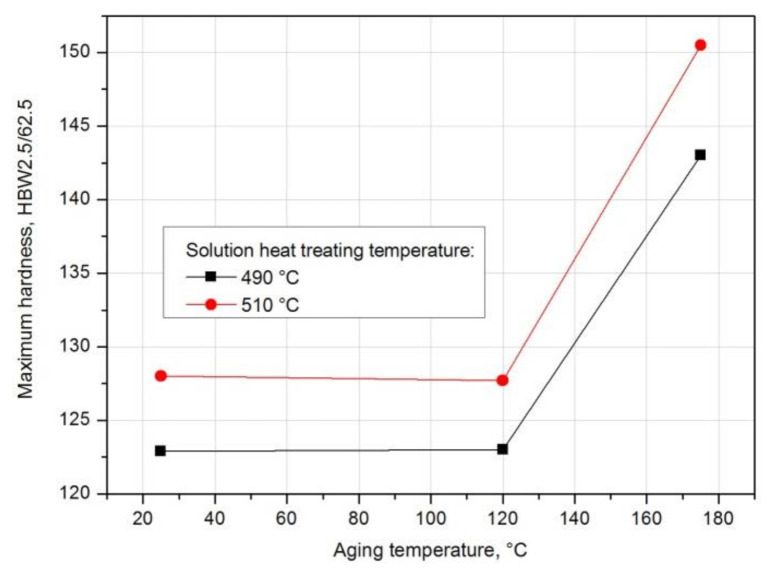
Effect of aging temperature and time on the change in the hardness of an alloy solution heat treated from 490 °C and aged naturally and artificially.

**Figure 20 materials-14-00089-f020:**
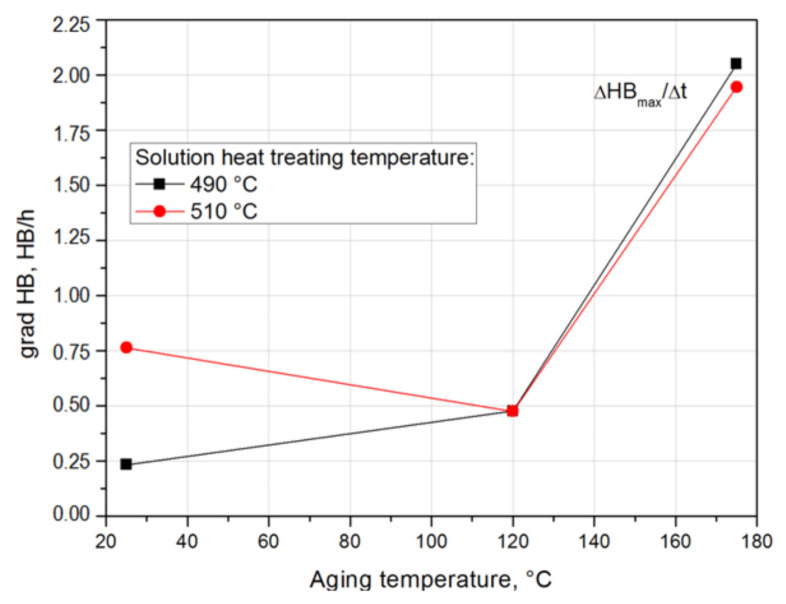
Effect of solution heat treatment and aging temperature on the rate of increase in hardness of the 2017A alloy during natural and artificial aging.

**Figure 21 materials-14-00089-f021:**
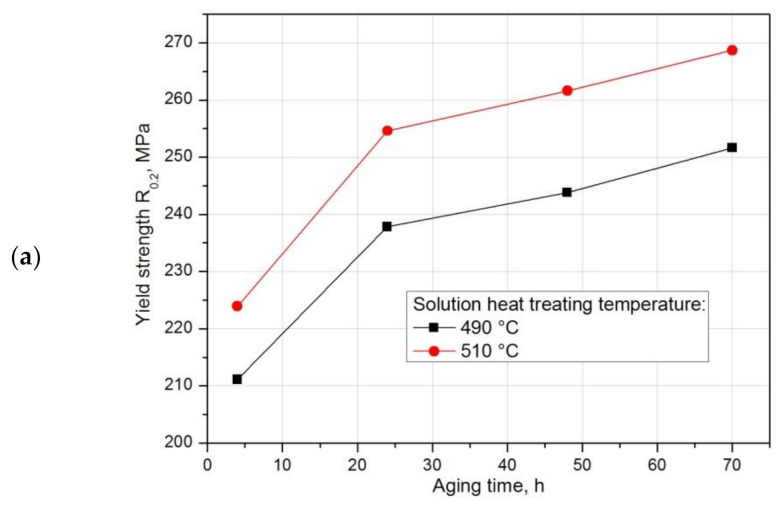
Influence of solution heat treatment temperature and natural aging time on (**a**) apparent yield strength R_0_._2_, (**b**) tensile strength R_m_, and (**c**) relative elongation A_5_ of the 2017Aalloy.

**Figure 22 materials-14-00089-f022:**
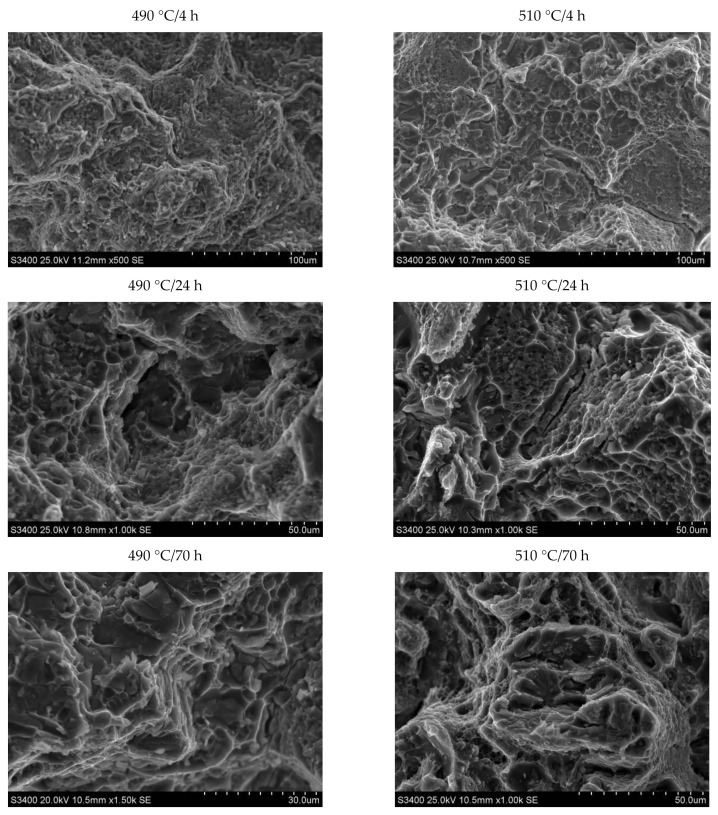
Images of fractures after precipitation strengthening and static tensile test.

**Figure 23 materials-14-00089-f023:**
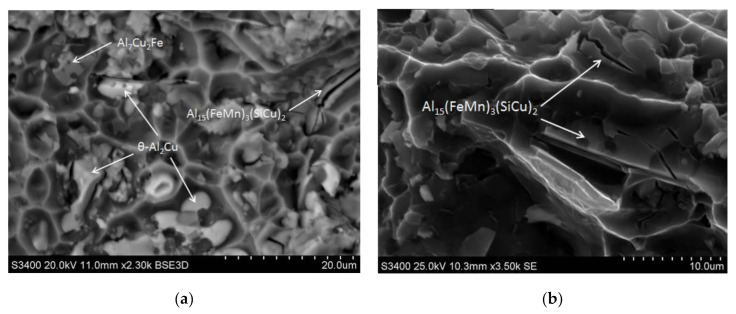
Images of fractures after precipitation strengthening and static tensile test, (**a**) the areas where the primary undissolved precipitates of the θ-Al_2_Cu phase occurred; (**b**) Al_7_Cu_2_Fe and α-Al_15_(FeMn)_3_(SiCu)_2_ particles defragmented as a result of tensile load.

**Table 1 materials-14-00089-t001:** Casting process parameters.

Temperature of Metal in Furnace (°C)	Temperature of Metal in Classifier (°C)	Amount of Cooling Water (L/min)	Casting Rate (mm/s)	Comments
740	690–700	30	3.5–4.0	Amount of water per one crystallizer. A total of 120 L/m for four crystallizers

**Table 2 materials-14-00089-t002:** Chemical composition of the 2017A aluminium alloy, weight %.

Alloy	Elements Content, wt %									
-	Si	Fe	Cu	Mn	Mg	Cr	Zn	Zr	Ti	Al
2017A alloy *	0.20–0.80	0.70	3.50–4.50	0.4–1.0	0.4–1.8	0.10	0.25	0.10	0.10	balance
2017A alloy **	0.49	0.22	4.01	0.56	0.72	0.063	0.20	0.17	0.077	balance

* Chemical composition of the 2017A alloy per the PN EN 573-3 standard ****** measured using a spectrometer on an alloy obtained from scrap using continuous casting.

**Table 3 materials-14-00089-t003:** Chemical composition of intermetallic phase particles (determined using SEM/EDS) in the as-cast 2017A alloy.

Number of Measured Particles	Suggested Type of Phases	%	Al	Si	Mn	Fe	Cu	Mg
35	α-Al_15_(FeMn)_3_(SiCu)_2_	wt	56.8–60.1	5.8–6.7	9.7–12.4	17.6–19.7	4.0–7.2	-
		at	70.6–73.2	6.9–8.0	5.9–7.4	10.3–11.8	2.7–3.8	-
24	Q-Al_5_Cu_2_Mg_8_Si_6_	wt	33.6–56.5	16.4–26.8	-	-	11.4–18.8	15.7–23.4
		at	37.0–59.7	16.7–27.8		-	5.1–8.8	18.5–28.1
25	Al_7_Cu_2_Fe	wt	61.5–64.4	-	-	8.9–9.4	25.0–29.1	-
		at	77.5–79.6	-	-	5.3–5.8	13.1–14.6	-
35	θ-Al_2_Cu	wt	47.0–47.8	-	-	-	52.2–53.0	-
		at	67.0–68.3	-	-	-	31.7–32.3	-
25	β-Mg_2_Si	wt	2.5–3.7	52.7–60.3	-	-	-	38.5–47.1
		at	2.6–3.6	47.3–49.5	-	-	-	47.7–52.7

**Table 4 materials-14-00089-t004:** Parameter values determined from aging curves necessary to determine hardness gradient grad HB.

Solution Heat Treatment Temperature, °C	HB_w_	HB_max_	ΔHB	Δt, h	Grad HB = ΔHB/Δt, HB/h
490	81.5	123.5	42	180	0.233
500	90.6	127.6	37	96	0.385
510	91.6	128.3	36.7	48	0.764

**Table 5 materials-14-00089-t005:** Characteristic temperature value of exothermic peaks 1–6 registered on the DSC curve (Figure 16).

Temperature, °C	Exothermal Peaks					
	1	2	3	4	5	6
Top of peak	70	138.6	184.3	238.0	269.9	329.6
onset	54.6	127.7	178.1	225.9	257.9	319.0
offset	81	141.3	188.4	247.6	282.3	345.9

**Table 6 materials-14-00089-t006:** Parameter values determined from aging curves necessary to determine hardness gradient grad HB.

Solution Heat Treatment Temperature, °C	Aging Temperature, °C	HB_w_	HB_max_	ΔHB	Δt, h	Grad HB = ΔHB/Δt, HB/h
490	25	81.5	123.5	42.0	180	0.233
	120	81.5	122	40.5	85	0.476
	175	81.5	143	61.5	30	2.05
510	25	91.6	128.3	36.7	48	0.746
	120	91.6	127.7	36.1	76	0.475
	175	91.6	150.0	58.4	30	1.946

**Table 7 materials-14-00089-t007:** Effect of solution heat treatment temperature and natural aging time on the strength and plastic properties of the 2017Aaluminium alloy.

Aging Time, h	Solution Temperature, °C	Yield Strength R_0_._2_, MPa	Tensile Strength R_m_, MPa	Elongation A_5_, %	Reduction Z, %
4	490	211.1	352.3	8.0	4.85
	510	223.9	416.4	16.3	15.7
24	490	237.8	362.6	6.35	4.45
	510	254.6	438.4	15.3	14.25
48	490	243.8	374.6	7.5	5.65
	510	261.6	444.4	14.9	13.4
70	490	251.65	398.35	8.85	7.3
	510	268.7	450.5	14.65	12.5

## Data Availability

Data is contained within the article or supplementary material. The data presented in this study are available in form of the final report from the project TECHMATSTRATEG. The report is open only for the authors of the article.

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
