# Peer review of "Microstructure and Properties of As-Cast and Heat-Treated 2017A Aluminium Alloy Obtained from Scrap Recycling"

_materials, 2020, doi:10.3390/ma14010089_

Round 1

Reviewer 1 Report

Dear authors,

You did an interesting work but the presentation of it does not satisfy. My remarks are as follows:

Abbreviations should be defined in parentheses the first time they appear in the abstract, main text, and in figure or table captions and used consistently thereafter.

In the Introduction there is a lack of literature overview about the current research status of aluminium alloy obtained from scrap recycling based on the available literature. Namely, introduction section does not provide sufficient background analysis regarding the production of Al alloy form scrap and therefore relevant references are missing.

Generally, it is not perfectly clear the exact number of samples tested using the various methods. Also, experiments presentation is a little bit confusing (it is not presented in complete chronological order).

P3 L116-118: How large/small were the samples for microscopic tests? Please specify in more detail how samples were grinded and polished. What were the temperature of the etching solution and time of etching?

P3 L122-126: Which radiation was used for XRD analysis, please specify conditions.

P5 L182-184: On how many samples the hardness was measured? How many hardness measurements were performed on each sample?

In Table 2, content of Al as base element should be stated (as balance).

In the caption of Table 3, wt.% should be deleted since in table atomic % were given as well. How many measurements of chemical composition for each phase were performed?

It would be desirable to compare the obtained results of the basic properties with those for Al alloy produced by melting pure elements (not by recycling).

Of the 41 references, only 7 of it are from 2015 and newer!

Extensive editing of English language and style is required.

Reviewer 2 Report

This paper is mainly investigated the phase composition and mechanical properties of 2017A aluminum alloys after casting and different heat treatments. It shows a good interest to reviewers at first sight because of recycling scrap aluminum alloys. However, there are some unsatisfied places and analysis, or even mistakes in this whole paper. It gives reviewers a not-well impression on your paper. The introduction and explanation of the treatment processes should be put in the second part (materials and method), and some information is repetitive. The structure of this paper is not very logical in science analysis. There are mistakes in this paper(some words marked with purple color in PDF), As following, there are some questions and suggestions about your paper.

(1) In the part “ Abstract “, you better write some important results and date to attract readers and reviewers.

(2) There should be more analysis of articles about recycling aluminum alloys which should be expressed in the part of the introduction, instead of the analysis of only aluminum alloys. And some dates or values you adopted in this part should be marked with related literature.

(3) There should be an analysis of changes in the structure of alloys under various external influences that modify the state of the material. For example, authors can use these articles: Ivanov, Yu.F., Gromov, V.Е., Konovalov, S.V., Zagulyaev, D.V., Petrikova, Е.А., Semin, А.P. Modification of structure and surface properties of hypoeutectic silumin by intense pulse electron beams (2018) Progress in Physics of Metals, 19 (2), pp. 195-222. DOI: 10.15407/ufm.19.02.195; Sarychev, V., Nevskii, S., Konovalov, S., Granovskii, A., Ivanov, Y., Gromov, V. Model of nanostructure formation in Al-Si alloy at electron beam treatment (2019) Materials Research Express, 6 (2), № 026540. DOI: 10.1088/2053-1591/aaec1f; Zaguliaev, D., Gromov, V., Rubannikova, Y., Konovalov, S., Ivanov, Y., Romanov, D., Semin, A. Structure and phase states modification of AL-11SI-2CU alloy processed by ion-plasma jet and pulsed electron beam (2020) Surface and Coatings Technology, 383, № 125246. DOI: 10.1016/j.surfcoat.2019.125246.

(4) If some literature proved that the temperature of solution heat treatment is approximate 490℃, you need mark related literature here (the part of 2.4 Heat treatment)

(5) the expression in the part of the statement above figure 3 is not very logical. It is better to introduce Figure 3 firstly and explain the reason for carrying out the precipitation strengthening process.

(6) the word “solutionise” is corrected?

(7) In figure 5 (b), what’s Φ? If it means the diameter of the ingot, the value is not accurate. There is no obvious microstructure for observing.

(8) the function of figures 5(b) and 6(a) is repetitive, can you change the picture of Figure 5 (b) or the expression?

(9) the expression “ Chinese characters…” should be marked with literature.

(10) In figure 7, typical EDS spectra of the analyzed particles should be marked with the alphabet.

(11) As for the analysis of figures 7 and 8 is not very well, you can give more information about it. What’s kind of causes leading to this phenomenon of phase composition?

(12) the microhardness of the 2017A alloy after casting is 98HB, while the value of it after solution heat treatment is decreasing (81.5, 91.6 HB). The mechanical property is not improved.

(13) In the third part of the analysis, you can divide into different parts, for example

(3.1)the microstructure of samples before any treatment)

(3.2)the analysis of microstructure and microhardness after the solution heat treatments

(3.3)the analysis of microstructure and microhardness after T4 and T6 treatments

(14) can you explain how to influence the kinetics of precipitation heat treatment temperature?

(15) you should summarize some information about the conclusions in the part of abstract.

Reviewer 3 Report

Present manuscript focuses on investigation of microstructure and mechanical properties 2017A Al alloy, obtained from scrap recycling, after casting, solution heat treatment in water and natural aging. The manuscript is well-written, and the language is concise and understandable. The research is original and well-organized and would be interesting for the readership of the Materials.

I have several observations:

1) Could you specify a type of ARL diffractometer and X-ray source, which are used in the study?

2) At the X-ray diffraction data (Figure 8, 10 and 15) some of indicated phases (Al7Cu2Fe, Q-Al5Cu2Mg8Si7, α-Al15(FeMn)3(SiCu2)2) are comparable with noise level. Thus, the conclusion about their presence was made solely on the basis of SEM/EDS?

Round 2

Reviewer 1 Report

Dear authors,

thank you for accepting my remarks. I hope you agree that they have contributed to the quality of the manuscript.

Best regards

Reviewer 2 Report

The article is now ready for publication in the journal.